# The Self Course: Lessons Learned from Students’ Weekly Questions

**DOI:** 10.3390/bs13070525

**Published:** 2023-06-22

**Authors:** Alain Morin

**Affiliations:** Department of Psychology, Mount Royal University, 4825 Richard Road S.W., Calgary, AB T3E 6K6, Canada; amorin@mtroyal.ca

**Keywords:** self-awareness, self-knowledge, mental time travel, self-regulation, self-recognition, self-esteem, culture, inner speech, Theory-of-Mind

## Abstract

In this paper, I tentatively answer 50 questions sampled from a pool of over 10,000 weekly questions formulated by students in a course entitled “The Self”. The questions pertain to various key topics related to self-processes, such as self-awareness, self-knowledge, self-regulation, self-talk, self-esteem, and self-regulation. The students’ weekly questions and their answers highlight what is currently known about the self. Answers to the student questions also allow for the identification of some recurrent lessons about the self. Some of these lessons include: all self-processes are interconnected (e.g., prospection depends on autobiography), self-terms must be properly defined (e.g., self-rumination and worry are not the same), inner speech plays an important role in self-processes, controversies are numerous (are animals self-aware?), measurement issues abound (e.g., self-recognition as an operationalization of self-awareness), deficits in some self-processes can have devastating effects (e.g., self-regulatory deficits may lead to financial problems), and there are lots of unknowns about the self (e.g., gender differences in Theory-of-Mind).

## 1. Introduction

The self has been a topic of interest, at least since human animals became aware of their own self, arguably over 60,000 years ago [1]. There is no universally accepted definition of the self, but a general agreement is that it is “… multidimentional in nature, made up of both conscious and unconscious layers, and is informed by observations of others” [2], p. 143. Here, the self is understood as all conceivable private and public aspects that make who a person is, including thoughts, emotions, goals, values, sensations, memories, traits, attitudes, physical attributes, behaviors, and skills Figure 2 in [3]. The main cognitive process that makes it possible for the self to apprehend itself and form an idea of itself (a self-concept) is self-reflection: the mental act of examining our self while being motivated by a healthy curiosity or interest in who we are [4]. There are many unknowns regarding the self, but a few solid empirical facts are emerging in the literature. One is that self-reflection and its negative counterpart, self-rumination, as well as several key self-processes, such as self-regulation, self-esteem, and self-knowledge, most likely rely on inner speech [5,6,7]—a verbal conversation we engage in with ourselves about ourselves. Another accepted view is that self-processes are interrelated in very complex ways [8,9]; therefore, if one mechanism is compromised (e.g., thinking about one’s past), others will also suffer (e.g., thinking about one’s future) [10].

The self has been studied within a wide array of disciplines, including psychology, neuroscience, philosophy, anthropology, sociology, and the arts. The self is also at the center of class discussions by students and instructors, who are eager to understand it better. As a matter of fact, the self can be a course subject in its own right, and some undergraduate and graduate university programs offer courses on the self.

This paper presents information about such a course, simply entitled “The Self”, created and taught by this author at the Department of Psychology at Mount Royal University in Calgary (Alberta, Canada). In what follows, I describe the course, its objectives, the modes of evaluation, and the main learning activities. Every week, 20 students are invited to read one or two key papers pertaining to a central aspect of the self and to produce three questions inspired by the target text(s). The instructor organizes these questions and brings them to class for discussion. Here, I examine the most representative weekly student questions from nine years of teaching this course and offer tentative answers to these questions. The questions and answers, I submit, greatly inform us on the most significant issues surrounding the self and allow for the identification of some recurrent messages about the self. Thus, the aim of this paper is to tentatively answer the students’ questions, as carried out in class, based on what we currently know about the self and extract key lessons from this exercise.

### 1.1. The Course

The Self course (a weekly 3-h class) was developed in 2014, following a decision to discontinue offering another course, Social Cognition, which already had a strong self-component, with topics including self-awareness, self-reflection, self-schemas, social comparison, fame, impression management, and attributions. The Self is a fourth-year undergraduate seminar-type course designed for third- and fourth-year students (20 students per section). Its format consists of critical discussions of representative scientific papers. The course is not lecture-based, where the instructor summarizes the key course material and students take notes based on the lectures. Rather, students are expected to read the assigned articles, make their own notes, come to class fully prepared to discuss the material, add their own ideas, and engage in a critical evaluation of the content.

Each week, students are invited to explore a key topic pertaining to the self. As mentioned earlier, they are asked to read one or two influential paper(s) on the topic and to produce three questions inspired by the readings. Also, each week, three to four students present empirical papers related to the target topic. In essence, the course discusses research results pertaining to the self from social-experimental, cultural, cognitive, psychopathological, comparative, developmental, and neuroscience perspectives; the nature and functions of the self are examined.

Topics and key readings are presented in Table 1. Abstracts for most papers are available in Appendix A. The selection of topics was based on the author’s comprehensive knowledge of the self, based on publishing in the area for more than 30 years (for an overview, see [9]). The selection criteria for the readings primarily include influential review papers likely to be of interest and comprehensible to students.

Some course objectives are as follows: (a) to define what the self is and to identify the main theoretical orientations, concepts, controversies, and researchers in this field; (b) to establish links between the self and other fields in psychology—e.g., personality, cognition, development, social, neuroscience, and comparative psychology; (c) to promote the ability to formulate solid and defendable positions on unresolved issues; (d) to examine a variety of methods used to study self-processes; (e) to develop students’ ability to critically read and examine research findings and theories presented in journal articles and book chapters; and (f) to increase students’ awareness of their own self.

The evaluation for the course consists of two exams, a term paper, class participation, two oral presentations of empirical articles pertaining to weekly topics, peer evaluations of the oral presentations, and most importantly, weekly questions mentioned earlier. The next section describes the last aspect of the evaluation in greater detail.

### 1.2. Weekly Questions: Instructions and Characteristics

As stated before, students are asked to formulate three discussion questions for each weekly key article(s) to be discussed in class—e.g., new research questions or new predictions not addressed in the target article(s). Only three questions are required, even if two articles are read. These questions are emailed to the instructor two days before class to allow time to read and organize them. The instructor prints a compilation of the questions, makes copies, and distributes them to students at the beginning of the class. Most of the time, themes emerge from the questions; that is, several students raise highly similar questions that can be put under a distinct heading. These themes are considered to be of prime importance and are addressed first in class. All other questions are randomly presented, and many of them are discussed until the end of class. About half of the weekly questions produced are actually addressed in class.

The total number of weekly questions produced by the students is 60 (again, 3 per student, 20 students enrolled per section). Since there are 11 topics covered (see Table 1), approximately 660 questions are submitted per semester. Sixteen sections were taught between winter 2014 and fall 2022, leading to the formulation of approximately 10,560 questions (the total number of weekly questions produced between 2014 and 2022 is approximative, as enrollment could vary slightly each semester, with either 19 or 21 enrolled students). The questions discussed in the next section are taken from this pool and reflect the most important issues that matter to students.

The selection criteria for the weekly questions to be presented in the next section included frequency (the emerging themes), pertinence in relation to the weekly readings, and perceived interest to readers. Questions that were poorly formulated or not relevant to the readings were not considered. In short, this author tried to select the most representative questions based on their pertinence to students and their importance to the research field on the self; clearly, this selection was subjective. An arbitrary number of 3 to 6 questions per topic was deemed manageable for this article; thus, 50 weekly questions in total are examined in the next section. Some questions have been rephrased for clarity.

### 1.3. Weekly Questions

(A) Self-awareness

(1) Can we be conscious without being self-aware?

Yes, very often, we do, feel, or think things without explicit knowledge of these contents [3]. Being conscious means being awake and responding to environmental stimuli [27], as when one is driving and engaging in all proper responses, such as changing lanes, maintaining speed, stopping at a red light, etc. One is immersed in the experience without reflecting on the experience itself (“I”, self-as-subject [28]). In contrast, being self-aware means becoming the object of one’s own attention and actively examining any salient aspects of the self [11]. To illustrate, the conscious driver suddenly realizes that (s)he is speeding—the driver is reflecting on his/her behavior (“Me”, self-as-object).

Whereas we can be conscious without being self-aware, but it does not work the other way around—we cannot be self-aware while unconscious. In unconsciousness (i.e., sleep, coma), we do not process information either from the environment or the self. (One exception could be lucid dreams when one is aware of dreaming while non-conscious [29]). This distinction between consciousness and self-awareness potentially explains the main difference between human and non-human animals’ inner experiences (see the next question A2).

(2) Are animals like my cat or dog self-aware?

This represents a highly controversial question, and undoubtedly, the answer depends on how self-awareness is defined. So many articles and books have been written on this subject (e.g., [30,31,32]), all informative but inconclusive in my opinion. Clearly, cats, dogs, horses, cows, pigs, birds, etc.— even fish and insects—are conscious in the sense of being awake, experiencing internal states, perceiving the external world, and adequately responding to environmental stimuli. To illustrate, a tiger must be conscious when hunting a deer, as this activity entails wakefulness, hunger, processing of visual, auditory, and olfactory information, and carefully approaching the target.

Is the tiger *self*-aware? The general agreement is that non-human animals must possess some level of self-awareness—at least bodily awareness—to navigate the physical world adaptatively. The tiger must process information about the position and location of its body to approach and attack a deer successfully. Birds need bodily awareness (kinesthetic information) to flock together in harmony and avoid bumping into trees [33]. The contentious issue is whether non-human animals are aware of more abstract self-aspects (private self-awareness [34]), such as their emotions, sensations, goals, needs, and memories. Here, it is important to recall the answer provided to question A1 above. The tiger may experience hunger (consciousness) without knowing about it (self-awareness). Pet owners regularly comment on the mental states of their pets, such as when saying “My cat is happy”, which is problematic. The cat may be experiencing positive internal states such as food satiation, but to claim that the cat is “happy” implies that it knows that it is feeling good, which is virtually impossible to verify. Furthermore, a state of happiness is not required to explain the cat’s behavior of (say) purring; satiation itself is sufficient.

When raising difficult questions that cannot readily be backed up with observable evidence, such as the current one, scientists must abide by Occam’s razor and select the simplest explanation available. Why is my dog all happy to see me coming back from work? A complicated, mentalistic account could be that the dog had been feeling lonely and anticipated your return with excitement. A simpler, more mechanistic answer could be that the dog came to associate its owner with food via classical conditioning and is behaviorally preparing to be fed. Unless proven wrong, the second option is the most scientifically viable. Concluding remark: There is no known valid and reliable way to confidently answer the question. The answer is likely to depend on the anthropomorphic inclinations of the person answering.

(3) Why is it that self-awareness is considered by some researchers as being highly beneficial, and by others as being detrimental to mental health?

Self-awareness does not represent a unitary construct. A classic distinction between self-reflection and self-rumination was introduced by Trapnell and Campbell in 1999 [4]. Self-reflection constitutes a healthy form of self-focus in which the person is intrigued by one’s own self and seeks self-discovery. This is a non-anxious form of philosophical curiosity toward oneself. Self-reflection is associated with positive outcomes, which include successful Theory-of-Mind (ToM; thinking out what others may be experiencing inside), self-regulation (i.e., becoming the person one wants to become), and self-knowledge (knowing who you really are). In contrast, self-rumination comprises self-doubts and negative self-appraisals that are repetitive and uncontrollable [35]. It is linked to negative outcomes such as anxiety, depression, ineffective self-regulation, and escape from the self via self-destructive behaviors. The self-ruminating person is said to be in a “self-absorbed” state that impedes ToM, as (s)he is uniquely interested in one’s miserable reality—not that of other persons [36]. It is impossible to make sense of the research results pertaining to self-awareness without keeping in mind the distinction between self-reflection and self-rumination.

(4) Are there specific personality traits associated with self-awareness?

Absolutely. However, here again, the distinction introduced above in the answer to question A3 prevails, as self-reflection and self-rumination are predictably associated with opposing traits. As a reminder, personality traits represent characteristics that are stable across time and situations [37]. In addition, it is customary to use the Big Five model of personality developed by MacCrae and Costa [38]. This model posits that we all differ in five universal dimensions: openness to experiences (O), conscientiousness (C), extroversion (E), agreeableness (A), and neuroticism (N, i.e., anxiety). In simple terms, individuals who are more self-reflective score relatively high on O (they are open to learning about themselves), C (they self-regulate more and are thus reliable), and A (they engage more often in ToM, which implies taking others’ perspectives, and are thus kinder and more respectful). Self-ruminative people score relatively low on O, C (poor self-regulation), and A (impeded ToM), but high on N (see [4]). To our knowledge, E is not meaningfully linked to either form of self-awareness.

(5) If a person was raised in total isolation, as in the case of Genie the feral child, how would this affect self-processes?

It is important to emphasize that the self represents a social construct that gradually develops via social comparison (gaining knowledge about our personal characteristics by comparing ourselves to others) and social feedback (others sharing with us their views of ourselves). Furthermore (as will be discussed with questions B2 and D3), inner speech is used in the form of internal verbal conversations that we address to ourselves to gain insight into who we are. Inner speech emerges from social speech [39], so without social interactions leading to the development of social language, inner speech would not develop.

Let us imagine the following fictive scenario. Michelle was born 50 years ago to parents who were mad scientists. They decided to raise her in total isolation right from birth to see what would happen. Thus, she was placed in an empty room only containing a toilet, shower, and bed, and was fed through a hole three times a day; no one ever talked to her.

What kinds of self-processes would Michelle have developed?

*None whatsoever*. Without social comparison and feedback or inner speech, Michelle would barely have a self and would be incapable of reflecting on it anyway. Since self-reflection constitutes a prerequisite for all other self-processes (see [13], Figure 1, as well as questions D1 and E1), Michelle would not engage in ToM (one cannot conceive of mental states in others without knowing about one’s own internal states), self-regulation (one cannot change the self if one is oblivious of one’s self), self-description, mental time travel, self-knowledge, self-concept, and ultimately self-esteem (one cannot evaluate one’s worth in the absence of information about one’s self). Thus, social isolation means the negation of self-processes. Social interactions are required for a self—and knowledge about it—to emerge.

(B) Mindfulness/self-knowledge

(1) I am confused. The article uses terms like mindfulness, self-reflection, and self-knowledge—what are the differences between them?

There are several differences between these terms. Mindfulness refers to a non-evaluative focus on the self in the present [14]; one is here, in the present, being aware of any salient self-aspects (e.g., bodily sensations, emotions) without asking questions about what is causing the experiences, what they could do or mean, etc.; just pure neutral self-observation. Carlson [14] further clarified that mindfulness involves “decentering”—watching oneself from a third-person perspective, thus adding a distance between the self and its inner experiences (personal communication, 20 January 2014). Other qualities of mindfulness include self-compassion, that is, observing the self in an accepting way without resisting anything that might be observed, nor trying to change experiences or reacting to them with thoughts or actions. Questionnaires assessing mindfulness as a disposition usually contain subscales representing sub-dimensions, such as present awareness and acceptance (e.g., [40]).

With self-reflection, the person actively (as opposed to passively) identifies, organizes, classifies, consolidates, questions, stores, and retrieves information pertaining to the self [7]. Although healthy [4], self-reflection allows critical self-evaluation and mental time travel (i.e., thinking about the past and the future, not just the present). Because of its more active and cognitive nature, self-reflection relies on inner speech, whereas mindfulness aims at silencing the inner voice [41]. Self-knowledge represents actual genuine information about oneself [14]; that is, information (e.g., about one’s personality traits) that can be corroborated by others. This is to be distinguished from self-concept, which is defined as who we think we are, irrespective of the accuracy of the information [13]. To illustrate, schizophrenic patients might see themselves as possessing grandiose qualities that most surrounding individuals would dismiss. The patients do exhibit a self-concept, but because it is unrealistic, they lack self-knowledge. Hence, the two terms must not be equated.

(2) The article claims that mindfulness increases self-knowledge—is that true?

To her credit, Carlson [14] presents creative and detailed research suggestions aimed at testing ways in which mindfulness might help overcome two main barriers to self-knowledge. These obstacles are informational barriers (things that are difficult to objectively apprehend, e.g., mannerisms) and motivational barriers (threatening things we do not want to know about, e.g., selfishness). Her main hypotheses are that mindfulness should lower these barriers by increasing our ability to sustain attention in the current moment, thus processing more information; in addition, the non-evaluative nature of mindfulness should reduce reactivity and defensiveness to ego-threatening information.

While the above hypotheses certainly sound reasonable, there are three key reasons to question the role of mindfulness in self-knowledge. (a) Because mindfulness exclusively focuses on the present, it ignores two important sources of self-knowledge: one’s recall of past significant personal events (autobiography) and one’s projection into the future (prospection). Remembering how we acted or felt in the past permits the detection of recurrent patterns that are informative (see [42]). For example, “Each time I was in a new intimate relationship, I always worried that my partner could be having an affair… perhaps I am jealous”. Additionally, imagining what we might become in the future allows for the identification of personal goals that shape who we indeed become [43]. (b) Because mindfulness is non-judgmental, it ignores the role of self-evaluation and self-criticism in the acquisition of self-information. For instance, one can learn that one is tardy by admitting lateness in several appointments. (c) As discussed above, mindfulness inhibits inner speech. The fact that inner speech plays a crucial role in many self-processes that leads to self-knowledge (see [13], Figure 1), such as self-reflection, self-description, and self-criticism, questions the link between mindfulness and self-knowledge.

In her paper, Carlson [14] laid the foundation for an entire research program on mindfulness and self-knowledge. Yet, intriguingly, there is no published research on this topic after 2013. Either she and others lost interest in this line of research, or attempts to test Carlson’s hypotheses failed. If the latter is accurate, the possible causes of unsupportive results could be those outlined above.

(3) Does self-knowledge fluctuate throughout the lifespan, or does one’s level of self-knowledge typically remain stable over time?

The main intrapersonal process leading to self-knowledge is self-reflection. Self-reflection is mostly conceived as a stable personality trait [44] and thus does not noticeably fluctuate over time. However, temporary states of self-focus induced by environmental stimuli (e.g., exposure to a mirror or an audience; [45]) may vary over time. Similarly, instances of interpersonal processes, such as social comparison and feedback, may be more or less frequent depending on one’s immediate social environment. The result of these somewhat mutable processes and events is a gradual gain in self-information that translates into self-knowledge. What is progressively acquired is permanently stored in memory and remains intact as we age. Actually, self-knowledge is remarkably resistant even in cases of dementia or brain damage [46]. 

(4) What are the possible consequences of a lack of self-knowledge?

Self-knowledge deficits can have very disagreeable effects. Imagine John, who *thinks* he likes the color yellow and decides to buy a yellow car—only to realize days after the purchase that he dislikes the car because of its color. Or picture Tomas, who goes ahead and gets a huge mortgage on a house he *thought* he would like—only to comprehend days or weeks after the banking transaction that he is now living in a house that he hates. The consequences of faulty self-understanding can be even more dramatic, such as when one pursues the “wrong” career path because of inadequate knowledge of one’s goals and skills, or when one keeps being involved in doomed intimate relationships because one does not know who the “right” partner should be.

A lack of self-knowledge basically signifies a distorted self-view, which is associated with bragging (as one overestimates one’s strengths), unrealistic choices, poor academic performance, and lower life satisfaction [14]. In contrast, knowing oneself well leads to realistic decision making pertaining to key aspects of one’s life, including the selection of compatible intimate partners and friends, education and career orientations that fit one’s goals, realistic choices about housing and geographic locations, and many more. And, as alluded to in the answer to question B2, self-knowledge facilitates self-regulation because it contains information about one’s goals, standards, and strategies to be used to shape one’s behavior in desired directions [43]. Thus, a lack of self-knowledge is likely to impede self-regulation.

(C) Mental time travel (MTT)

(1) I am confused. What are the differences between MTT, imagination, mind wandering, daydreaming, and planning?

These terms all potentially imply past or future thinking but differ in important ways. MTT refers to cognitively remembering one’s own past (episodic memory; autobiography) and imagining one’s own future (prospection; episodic future thinking). By definition, MTT is always about oneself. Any past/future thinking that is not about the self is not MTT. Take imagination as a case in point, which constitutes the faculty of forming new ideas, images, or concepts of something not present to the senses [47]. If one imagines something that pertains to one’s life in the future (e.g., how an upcoming class presentation will go) or in the past (e.g., swimming for the first time), one is essentially engaged in prospection and autobiography, respectively. If one is imagining a totally fictive scenario not directly related to oneself (e.g., the first contact between humans and an alien race), or imagining something self-related in the present (e.g., what food to eat now), these exercises in imagination are not MTT. Mind wandering and daydreaming (largely synonyms) constitute arbitrary thoughts experienced when engaged in attention-demanding tasks [48], such as when one catches oneself thinking about something unrelated to the book one is reading. Since both may occur in the present and be non-self in nature, they must not be equated with MTT. Finally, because making plans for something [49] is unavoidably about the future and most usually about ourselves, this suggests that planning represents a special case of prospection. However, if planning is not associated with the self, as when a travel agent plans an itinerary for a client, it is not prospection.

(2) What happens when someone loses the ability to think about his/her own past?

The loss of access to one’s memories following a brain insult or dementia reliably results in prospection deficits [50]. In other words, prospection relies heavily on episodic memory because the latter provides building blocks from which episodic future thoughts are constructed (e.g., [10,51]). To illustrate, one can foresee what an upcoming trip will look like based on memories of past trips; without these memories, it would be very hard to imagine what any future trip could be like—although one could still use semantic information gathered in travel books and such.

The evidence of a solid causal link between autobiography and prospection includes the observation that patients characterized by poor episodic memory (as in depression and schizophrenia) exhibit difficulty in imagining their future [15], and common brain activations are recorded when participants engage in autobiographical and prospective tasks [10].

(3) What is the purpose of episodic future thinking and how does it benefit us?

Prospection serves an important survival function as it allows us to set future goals, plan, self-regulate, and ultimately reach these goals, thus becoming who we want to become. In short, episodic future thinking guides our behaviors in adaptative ways. For example, John could envision himself suffering from lung cancer because of smoking (prospection), so he decides to quit smoking now (self-regulation). Or, Martin might see himself as a successful lawyer in 10 years and enrolls in a law school to obtain a degree.

(4) For people who suffer from depression, are episodic future thoughts more positive, negative, or neutral in nature? What about in anxious people?

Research shows that when depressed individuals think about their future, they tend to report fewer details in imagined positive experiences compared to neutral or negative ones. Specifically, simulated future events exhibit less detail/vividness, less use of mental imagery, less use of first-person perspective, and less plausibility/perceived likelihood of occurrence. This suggests a reduced anticipation of a pleasurable future (e.g., [52]), which is obviously consistent with depression. Moustafa and colleagues ([53], p. 7) propose that depressed people might struggle to imagine a good future because they struggle to remember a good past. On the other hand, the same research group observed that anxious people have greater difficulty in generating positive and negative future events, possibly as a means of controlling anxiety via reduced anticipation of future events in general.

(5) Why is it that healthy people tend to be optimistic when they imagine their future?

Human beings are remarkably optimistic, and this colors their imagining of the future. The main reason for this is that viewing our future as good and pleasant motivates us to keep going and working at attaining our future goals. Simply put, if we were pessimistic about our future, we would stop acting, or at least we would do less, thus harming our motivation to evolve and succeed (see the answer to question C3.)

(D) Theory-of-Mind (ToM)

(1) Two weeks ago we studied self-awareness. What is the connection between self-awareness and ToM?

The most accepted view is that self-reflection (the healthy form of self-awareness) causally leads to (precedes) ToM. Self-reflection allows one to apprehend one’s inner experiences and to imagine the existence of similar experiences in others. This refers to the simulation theory: people mentally simulate what others might be internally experiencing by imagining what types of experiences they might have themselves if in a comparable situation [54]. Recall, however, that self-rumination impedes ToM as one is too absorbed in one’s (negative) experiences to bother considering others’ experiences (see the answer to question A3.)

Here is an example of the simulation theory: Robert goes to the dentist for a root canal procedure; as the dentist proceeds, Robert is aware of several perceptions (most being unpleasant), such as the needle penetrating his gum, the taste of the freezing agent being administered, the vibrations produced by the drill, the overall post-procedure sensation, and so forth (self-reflection). Several weeks later, Robert meets with Ruth, who tells him that she just got back from the dentist. Robert expresses his sympathies and tells her that he knows exactly how she feels (ToM) because he too went through a similar experience not long ago (simulation). Then, Robert perhaps offers Ruth some ice cream to make her feel better (helping behavior).

Empirical evidence in support of the simulation theory is overwhelming: (a) the more people are effective at self-reflection, the better they are at reading others’ minds [55]; (b) self-reflection interventions in schizophrenic patients precede ToM improvement [56]; (c) there is an important overlap in brain activity when participants work on self-reflection and Theory-of-Mind tasks [57]; (d) traumatic brain injury patients who exhibit self-reflection deficits also show ToM impairment [58]. Note that despite these compelling observations, some argue (e.g., [59]) that ToM leads to self-awareness, not the other way around. Another view is that self-reflection is required for the emergence of ToM but not for its online use. That is, once fully developed based on introspection on one’s own mental states, ToM could take a life of its own and not necessitate constant introspection when reflecting on others’ mental states [60].

(2) What would happen if we would all lose our ability to engage in ToM overnight?

The answer to this question emphasizes the formidable importance of ToM. One could argue that without ToM, the end of the world would ensue within a few weeks or months. ToM allows for the development of at least some understanding of others’ reality; without such an understanding, the world would collapse. ToM represents the ability to attribute mental states (e.g., goals, intentions, beliefs, desires, thoughts, and feelings) to others [17]. It serves several key functions, including smooth social interactions, prediction of others’ behavior, helping behavior and cooperation based on empathy toward others, detection of, and engagement in, deception to gain an advantage over others in a competitive environment, cheating when necessary, and avoidance of others when threatening intentions are perceived [61]. Without ToM, we could not effectively communicate with one another [62], as a basic knowledge of the communicator’s perspective and implicit underlying information about what they mean are required. In short, ToM is required for the survival of the species.

(3) Is language (including inner speech) important for ToM?

Several researchers maintain that ToM depends on the acquisition and use of language (e.g., [63,64]). Indeed, both phenomena emerge at about the same time, between three and five years of age. Language is associated with ToM because early social interactions—which are instrumental in ToM development [65]—involve verbal conversations between the child and caregivers [66]. During such conversations, caregivers may disclose their own inner experiences and ask about others’ mental states (e.g., “How do you think your friend feels about this?”), thereby motivating the child to adopt others’ perspectives. In addition, language allows for the development and use of a complex set of verbal labels (vocabulary) about our own and others’ feelings, thoughts, desires, beliefs, etc. This labeling of mental states facilitates their identification and differentiation [6]. It is postulated that conversations and label use are internalized via inner speech [39] and that we continue to internally verbalize about (our and) others’ mental states in adulthood (see [67]). To illustrate, think of the many occasions when we internally verbalize to ourselves statements like “Did I hurt her feelings?”, “Why did he say that?”, “Tomas lost his cat and probably feels sad”, and so forth. In support of the above, individuals in the autism spectrum who are known to exhibit ToM deficits also show underuse of inner speech (e.g., [68]).

(4) How would ToM in a narcissist or psychopath differ from the normal person?

At first glance, one would assume that both narcissistic and psychopathic individuals lack ToM skills, accounting for their absence of guilt or remorse following acts that hurt other people. A closer inspection suggests that these individuals may possess one form of ToM and may be deficient in a second type of ToM. This distinction is based on the notion of cold and hot cognitions (see [69]). “Cold” cognitive ToM refers to calculated thinking about others’ mental states, sometimes at the service of deception. For instance, a psychopath could wonder when a target person will leave the house in order to rob its content. Cold ToM is known to recruit higher executive functions relying on cortical areas. “Hot” ToM, on the other hand, is involved when we emotionally connect with the feelings of other persons, allowing us to exhibit empathy. What the psychopath is unlikely to do is think about the distress the robbed person will experience. Indeed, what prevents most of us from hurting others for personal gain is the empathy part of ToM. Hot ToM is associated with the activation of subcortical brain areas, subserving the processing of emotional information.

(5) One of the articles mentions that females are better than males at ToM—why is that?

The first thing that must be kept in mind is that females’ advantage at ToM compared to males is statistically small—it would be wrong to claim that females “excel” at ToM and males show “poor” ToM performance. That being said, these gender differences might stem from socialization practices (see [70]), where females are typically encouraged to focus more on their emotions and to express them more freely. To simplify, it seems that girls are taught to “internalize” by analyzing their inner experiences to a greater extent, as well as sharing these states with others. Instead, boys tend to “externalize” more via a showy display of their internal states (e.g., aggression in sports). Females’ greater introspective disposition essentially amounts to more self-reflection, which in turn possibly means more simulation of others’ mental states, hence the ToM advantage (recall the simulation view discussed in question D1). Another complementary possibility is that most of the above is linguistic in nature (i.e., interpersonal conversations) and gradually becomes internalized via inner speech, which is known to positively sustain self-reflection and ToM (see question D3). Arguably, females tend to be more verbal than males (see [71]).

(6) Do animals possess ToM?

As for question A2 regarding animal self-awareness, this is a contentious issue. Let us start by acknowledging that, similarly to self-awareness, ToM does not represent a unitary construct and most likely involves levels, so it is not an all-or-none phenomenon. Let us also agree that animals must possess at least some primitive form of ToM; otherwise, they would not survive (question D2). A basic understanding of other animals in terms of behavior prediction (i.e., deception and cooperation) is required for survival. Thus, the question rather is: up to what point do animals exhibit ToM abilities?

Some researchers impute almost human-like ToM to primates (e.g., [72]) based on anecdotical observations made in the wild. Designing convincing experiments under controlled conditions is remarkably difficult (see [73,74]), as non-mentalistic interpretations can often explain what seems to represent instances of ToM [75]. In their literature review, Call and Tomasello [76] propose that chimpanzees can understand goals, intentions, perceptions, and knowledge in other conspecifics, but not their beliefs. In a typical experiment on intention understanding, chimps observe a human experimenter trying to turn on a light with his head as his hands are occupied holding a blanket. The animal reacts to this not by imitating the experimenter’s behavior (i.e., turning on a light with its head), but instead by imitating the *intention* behind the action, which is to turn on the light with its hands. Povinelli and Vonk [77] suggest that chimpanzees form mental concepts of visible, concrete objects in their environments (e.g., trees, facial expressions, and other chimpanzees), but not about unobservable things such as God, gravity, or love. When engaged in ToM, chimpanzees would reason about the statistical regularities that exist among certain events and the behavior, postures, and head movements of others (behavioral abstractions), but not about others’ covert mental states.

The type of ToM possessed by other non-primate animals is unknown. One can assume that at least an online, quick—as opposed to an offline, deliberate—version exists for survival purposes. Pet owners often utter things such as “My dog knows when I am sad”, which would be a clear case of ToM—empathy, to be more precise. Multiple animal behaviors may appear to imply ToM on the surface, but in most cases, a non-mentalistic explanation can be offered. In the example above, it is very likely that the dog has gradually learned to walk toward its owner when (s)he is crying because such an approach response was reinforced in the past (that is, owner cries > dog approaches > owner pets dog > dog is reinforced). Thus, the apparent instances of ToM in non-primate animals can be understood in terms of classical and operant conditioning. Let us recall Occam’s razor discussed in the answer to question A2; that is, always select the simplest explanation obtainable.

(E) Self-regulation

(1) Four weeks ago we studied self-awareness. What is the connection between self-awareness and self-regulation?

As defined in the reading [18], self-regulation consists of altering one’s behavior, changing one’s mood, selecting responses from options, and filtering irrelevant information. In essence, self-regulation refers to the control of one’s behavior, emotions, and thoughts in pursuit of long-term goals. A general principle is that we cannot change, alter, select, and filter things about ourselves we are oblivious of. Thus, self-awareness (specifically, self-reflection) represents a prerequisite to self-regulation, as one must be cognizant of what self-aspects need to be modified before effective cognitive-behavioral control can occur (e.g., [78]). Most importantly, self-regulation heavily relies on standards as well as comparisons between the ”real” self (who one currently is) and the “ideal” self (who one wants to become) [79]. Such comparisons require self-reflection. The importance of self-reflection in self-regulation is emphasized by Bandura [80], who views the former as the first step in the latter. Note that basic animal self-control (see question E4 below), such as when a dog refrains from biting another dog, most likely does not require self-reflection (see [81]) and instead relies on instinctual responses.

This is the third of several instances (see [13], Figure 1) where self-reflection causally precedes another self-related process: recall self-knowledge and ToM. Consistent with the above discussion, inner speech, which is active during self-reflection [6], is also important during self-regulation, suggesting the following causal chain: self-reflection > inner speech > self-regulation. A large body of research based on Vygotsky’s ideas [39] shows that private speech (out loud self-directed speech in children) causally influences several self-regulatory outcomes [20,82]. For example, the performance of children on the Tower of London task (a measure of planning, which represents an important part of self-regulation) is lower when private speech is blocked using articulatory suppression [83]. Articulatory suppression is achieved by having volunteers repeat a word over and over (or counting backward from 100), thus interfering with the ability to emit self-verbalizations. Also, employing articulatory suppression, Tullett and Inzlicht [84] observed self-control deficits in adults on a “go/no-go” task. Meichenbaum and Goodman [85] designed a self-instructional training procedure aimed at developing inner speech use and showed a reduction in impulsive behavior in children. In addition, Duncan and Cheyne [86] observed more private speech produced by young adults when working on a difficult task as opposed to an easy one; note that problem-solving also constitutes an important part of self-regulation.

(2) What happens when we fail at self-regulation?

The consequences of self-regulation failures are not trivial and echo those of self-knowledge deficits. Research shows that self-regulation failure is associated with a host of negative outcomes, such as violent behavior and anger issues, addiction, poor health (e.g., obesity), impulse buying and financial problems, poor decision making (e.g., unsafe sex), under-achievement, relational problems, and low self-esteem (see [87]). Opposite outcomes are associated with effective self-regulation. Vivid examples of self-regulation failures include spending large amounts of money purchasing pricy items online, having unprotected sex with one’s best friend’s husband/wife, and physically harming another person in a conflict situation.

(3) Are there psychological disorders associated with self-regulatory problems?

This question nicely complements the previous topic addressed in question E2. Indeed, multiple psychological disorders are linked to self-regulatory difficulties. A case in point is Attention Deficit/Hyperactivity Disorder (ADHD) [88] because attention, like self-reflection, represents a prerequisite to self-regulation. The same applies to addiction, defined as the inability to prevent oneself from continuing to use it using [89], as well as binge eating. Additional examples are depression (self-rumination interferes with self-regulation), bipolar mood disorder (during a manic state, individuals experience problems regulating their extremely positive emotions), and Borderline Personality Disorder (BPD). In the last case, the person experiences intense mood swings and feels uncertainty about how to perceive oneself. Feelings toward others can change quickly and swing from extreme closeness to extreme dislike. These changing feelings can lead to unstable relationships and emotional pain. Thus, a significant part of BPD is a difficulty in regulating one’s emotions [90].

(4) I wonder if there is a difference between self-regulation and self-control.

There is no universally accepted distinction in the scientific community between self-regulation and self-control (but see [91]); both terms are frequently used interchangeably. However, for clarity purposes, one can conceive of self-regulation as a broader process comprising multiple narrower self-control efforts. Self-regulation is long-term (e.g., days, weeks, months, or years) and entails executive functions, such as working memory (i.e., inner speech), planning, and decision making. An example could be obtaining a university degree, which takes years and requires foresight, discipline, organization, and perseverance. Students must make many crucial decisions, such as what courses to take and in what order, how and when to study and work on assignments, how and when to pay tuition fees, and so forth.

Instead, self-control is short-term (e.g., minutes or hours) and implies a delay of gratification (“Chocolate only after dinner”) and resisting temptation (“Only one piece of chocolate”) in the pursuit of long-term goals (e.g., weight management). Students who force themselves to keep studying instead of watching a television show (“I’ll watch the show only after 30 min of studying”) are effectively engaged in self-control. The same students will need to repeatedly exert self-control efforts throughout the larger self-regulatory process of obtaining the degree, such as when declining to go to a party to write an essay instead or saving money here and there to afford tuition.

(F) Inner speech

(1) There are so many terms used to designate inner speech, such as private speech, self-talk, internal dialogue, speech-for-self, or propositional thought—are there differences between these terms?

The use of different expressions to designate the same construct is deplorable as it adds confusion and complicates literature searches. That is, one must include *all* existing terms such as inner speech, propositional thought, self-verbalization, self-directed speech, self-statements, silent verbal thinking, phonological loop, or subvocal speech in search engines to capture what is known about the phenomenon. The umbrella term “self-talk” constitutes the activity of talking to oneself out loud or in silence [92]. The latter is usually named “inner speech”, defined as the capacity to produce language silently in one’s head [93]. However, there are some differences between these labels. For example, “internal dialogue” refers to self-directed speech made up of back-and-forth comments to oneself, as when one converses with another person (see [94]), whereas “internal monologue” signifies self-talk in which one talks to oneself as one person. “Private speech” describes self-talk emitted out loud by children in social situations without preoccupation with being understood by others [20]. Note that Piaget [95] favored the term “egocentric speech”, suggesting that private speech represents an immature manifestation of cognition. This idea was in sharp contrast to Vygotsky’s view [39] that private (and later inner) speech plays an important self-regulatory function (see questions E1 and F3).

(2) I sometimes talk to myself out loud—is this normal or am I crazy?

You are not crazy! As long as the self-talk is emitted when alone, it is perfectly normal and actually healthy as it serves important cognitive functions (see question F3 below). Private speech in adults differs from that of children in that it is not carried out in the presence of other persons, although one can be caught talking to oneself out loud by an inadvertent observer. Adult private speech has long been negatively perceived, most likely because of its association with schizophrenics’ or alcoholics’ incoherent self-verbalizations in public. That adult private speech is “crazy” is a myth, as demonstrated by Duncan and Cheyne [96], who developed a private speech scale and administered it to a large sample of healthy young adults. They found that a substantial amount of private speech was reported by the participants. Unlike the developmental trajectory of private speech in children (see [97]), little is known about several issues such as individual differences in adult private speech. What is known is that the frequency of private speech reliably increases when people are confronted with new or difficult tasks [98] (e.g., “Where am I supposed to plug this wire” when installing a novel and complicated sound system). Incidentally, Vygotsky [39] maintained that once well internalized as an inner speech during adolescence, private speech ceases to exist; however, the evidence presented above does not support this assertion.

(3) What would happen if I had a stroke and lost my inner voice?

The void felt by the absence of our inner voice would be dramatic. The question above is precisely what happened to Jill Bolte Taylor, as detailed in her 2006 book [99] and analyzed in Morin [100]. Jill suffered from a left hemispheric stroke caused by a congenital arteriovenous malformation, which led to a loss of inner speech. Several of her self-processes were altered—for instance, deficits in corporeal awareness, sense of individuality, retrieval of autobiographical memories, and self-conscious emotions. This is consistent with the view that inner speech serves important self-reflective functions [6], as already expressed multiple times in this paper.

Another way to answer the question consists of looking at what people report typically talking to themselves about and imagining a psychological world without these contents. In our work (e.g., [9,101]), we use an open-format measure aimed at collecting self-reported instances of naturally occurring inner speech in response to three probes: I talk to myself about (topics), when (during activities), and in order to (functions). Volunteers (students) freely list their recalled inner speech topics and functions, as presented in Figure 1. Table 2 displays less frequently reported contents.

Now, picture yourself *not* being able to talk to yourself about all these multiple themes and for such various reasons: a mind without its inner voice would be very silent and lonely indeed, suffering from severe self-reflective, self-regulatory, and cognitive deficits.

(4) How often do we talk to ourselves? Is it possible to have no inner voice?

It is very difficult to estimate inner speech frequency. Hurlburt and colleague (e.g., [102]) used the descriptive experience sampling (DES), which relies on a beeper that randomly cues participants to report whatever inner experiences they had just before the beep. Participants reported experiencing inner speech about 20% of the time they were probed, with important individual differences ranging from 0% (no inner speech) to 75%. This team reported a much higher inner speech frequency (over 70%) using a self-report scale they developed (the Nevada Inner Experience Questionnaire [103]). One explanation offered is that the DES method entails training and practice that is likely to increase participants’ understanding of what inner speech is and is not, which the self-report approach does not, leading to an overestimation by volunteers using the latter.

Thus, it appears that we do talk to ourselves a lot. Note that, as seen above, some people report experiencing no inner speech at all. Do these individuals have a silent mind as alluded to in the answer to question F3? Perhaps, but more probably, they *do* experience inner speech but do not know about it. This “silent mind” issue is currently hotly debated in several blogs on the internet.

(5) Do babies or animals have inner speech?

Few self-related questions can be answered with a high level of scientific confidence. Exceptionally, this question can receive a straightforward and assured answer: no. It is an empirically well-established fact that social speech comes first, followed by private speech, and then inner speech. In other words, self-directed speech becomes gradually internalized, from outer/interpersonal to inner/intrapersonal [39]. One cannot possibly exhibit inner speech prior to acquiring social speech. Since babies and animals do not have social speech, they do not have inner speech.

(G) Self and brain/traumatic brain injury (TBI)

(1) What are the main causes of TBI?

TBI results from a bump, blow, or jolt to the head, or a penetrating injury (e.g., a gunshot) to the head. This is important because it means that degenerative brain diseases, such as Alzheimer’s or dementia, although they may lead to similar symptoms (including anosognosia; see question G2 below), are technically not TBI. Men and older adults are more at risk of suffering from TBI. According to the Centers for Disease Control and Prevention [104], the most frequent causes of TBI are violence (including child-, domestic-, and war-related), transportation and construction accidents, sports, and falls.

(2) Are patients suffering from TBI in denial regarding their deficits, or are they truly unaware of them?

They are genuinely unaware of their deficits, which is called “anosognosia” [21]. Anosognosia encompasses several forms of sensory, motor, perceptual, cognitive, and/or emotional disturbances [105]. Patients with TBI who claims they can tie their shoes or run without difficulty truly believe so to the dismay of close ones who (literally) know better. TBI victims are *not* in denial (i.e., being cognizant of deficits but refusing to admit their existence) because they do not exhibit (or report) emotional distress or depression in light of their deficits. Instead, patients display a flat affect inconsistent with a realistic apprehension of the painful situation.

(3) Can we use self-report questionnaires to assess anosognosia in patients with TBI?

This is not advisable [21]. By definition, TBI patients lack awareness of their condition; thus, their insight into it will likely be distorted and unreliable (i.e., overestimation of their skills and underestimation of deficits). Basically, self-report questionnaires rely on self-reflection, which is impeded by anosognosia. Better methods for assessing anosognosia (see [106]) include objective measures of actual performance and asking close ones (e.g., family members and therapists) to rate patients’ performance. In the second case, a lack of convergence between self and others’ performance ratings is likely to be observed and signifies the presence of anosognosia.

(4) Why is it important to reduce anosognosia after TBI and how is it done?

It is critical to reduce anosognosia as much as possible for two main reasons. (a) The less patients are aware of the severity of their deficits, the less they will see a need for rehabilitation, and thus therapy compliance will be compromised. This, in turn, reduces the likelihood of physical and cognitive improvements post-TBI. (b) TBI patients lack insight into their condition and may expose themselves to injuries when engaging in behaviors that they cannot perform adequately. To illustrate, a patient might decide to go for a walk despite locomotion deficits and falls. In essence, anosognosia increases the chances of injuries—hence, the need to address it in therapy. Therapy consists of providing gradual and objective feedback to the patients regarding what they cannot do as well as before the trauma [107]. The feedback can be verbal or visual (i.e., using video recordings), and must be gentle and supportive to avoid emotional distress. Inviting patients to practice motor skills in a virtual reality environment represents an excellent approach [108] because of the absence of risk if performance fails.

(H) Self-recognition

(1) One article sees self-recognition as being a good measure of self-awareness, and the other one disagrees. Can you explain?

The significance of mirror self-recognition for self-awareness has been debated in the literature since Gordon Gallup Jr. published his first report of successful self-recognition in chimpanzees in the sixties [109]. The supporters of Gallup’s view (to be described below) include Julian Keenan, James Anderson, Alvaro Pascual-Leone, Lori Marino, and Mark Wheeler. Among critics, one can find Robert Mitchell, Daniel Povinelli, Cecilia Heyes, and this author (see [23,110] for a summary of the debate).

Gallup’s interpretation of successful self-recognition is as follows (see [111]). Exhibiting self-directed behaviors in front of a mirror suggests that the organism can take itself as the object of its own attention; the ability to self-focus represents a well-established constituent of self-awareness [112]. Furthermore, recognizing oneself in front of a mirror assumes pre-existing “self-cognition” (i.e., a self-concept), and thus self-awareness. More specifically, Gallup and colleagues maintain that self-recognition entails knowledge of one’s mental states, such as emotions, thoughts, motives, beliefs, and so forth. The main point made by opponents of this rationale is that the only prerequisite for self-recognition is information about one’s body—a non-mentalistic dimension of the self. All the self-recognizing organism needs is a kinesthetic representation of its own physical self, which is matched with the image seen in the mirror. (Additional evidence against Gallup’s view will be discussed in upcoming questions.)

Gallup’s interpretation has been taken as absolute truth by many, to the point that introductory or developmental psychology textbooks uncritically claim (for example) that children become self-aware once they become capable of self-recognition. The problem goes further, as this interpretation is now being used to question the idea that inner speech is involved in self-awareness. To illustrate, Kohda and colleagues [113] recently presented what they claim is evidence of self-recognition in fish. The authors’ construal of the evidence implies that since there is self-recognition in fish, they are self-aware; fish do not use inner speech, and thus inner speech is not important for self-awareness. Clearly, such reasoning ignores the large body of evidence in favor of the self-reflective functions of inner speech (notably, see question F3) and directly compares human and fish cognition, which is problematic [114].

(2) Is it possible for an organism to have self-awareness and yet fail the mirror test?

Absolutely. Blind individuals obviously will not be able to self-recognize in front of a mirror despite intact self-processes, such as self-reflection, self-knowledge, self-regulation, and MTT. Prosopagnosia represents a neurological condition characterized by the inability to recognize faces, including one’s own [115].Patients suffering from prosopagnosia fail the mirror test (i.e., touching a mark on one’s face in front of a mirror) despite intact self-awareness. Note that the opposite is also true: it is possible to achieve self-recognition notwithstanding severe self-awareness deficits (see [23]), such as in Down syndrome, autism, schizophrenia, and any other imaginable mental disorders for that matter. Self-recognition is remarkably resistant to disorders, diseases, and dysfunctions, and may be affected only in extreme cases of dementia. Indeed, self-recognition is established very early in development (between 18 and 24 months of age; [116]) and is one of the last functions to go. The above further casts doubt on Gallup’s interpretation of self-recognition for self-awareness: how can these two processes be so intimately connected when there is clear evidence that the former can be observed in the absence of the latter, and vice versa?

(3) What animals can recognize themselves in a mirror?

It is not an overstatement to state that basically all living creatures have been tested for self-recognition. To date, non-human animals who have passed the mark test are chimpanzees, orangutans, bonobos, elephants, dolphins, magpies, jackdaws, and fish (see [117]). All other creatures have failed the test and/or do not emit self-directed behaviors in front of a mirror. Note that some animals seem unsuccessful at self-recognition, not because they lack the ability but because they react to reflecting surfaces, unlike most other animals. For example, gorillas avoid eye contact during social interactions, and thus dislike looking at their own reflection in a mirror [118]. And no, cats and dogs do not recognize themselves in a mirror [119].

(4) Are there other measures of self-awareness in animals besides self-recognition?

One promising research avenue is the study of metacognition in animals [120]. Metacognition represents knowledge about one’s own thinking processes—thinking about thinking [121]. One manifestation of metacognition is uncertainty responding (UR): knowing that one does not know or might not remember something. When uncertain about something, we tend to decline to respond or differ in responding to seek more information. This indicates insight into our own thinking. For example, one could be asked when a specific event occurred; if one is uncertain about the date, the response could be delayed to check the information. Testing UR in animals involves asking subjects to discriminate between high and low tones (in auditory tasks) or high- and low-density images (in visual tasks); subjects must decide if a stimulus is higher (or lower) than another. They are given the option of declining or delaying trials when the discrimination becomes too difficult, indicating UR when they do so (see [122]). Apes, dolphins, and some monkeys seem to have metacognitive abilities.

It is important to keep in mind that metacognition represents only one facet of self-awareness; the demonstration of its presence in some animals does not mean that they possess other self-processes, such as additional aspects of self-reflection, or MTT and self-knowledge.

(I) Negative self-focus/psychopathology

(1) Is there a difference between rumination and worry?

Both rumination and worry represent repetitive, uncontrollable, and negative thoughts. An important difference is that rumination is always about the self (hence “*self*-rumination”), especially one’s past or expected failures, whereas worry may be non-self as when one worries about another person’s poor health or a hurricane devastating some part of the earth. One could say that all rumination is worry, but not the other way around: some worrying is not self-rumination.

(2) The article mentions that women tend to ruminate more than males—why?

The possible answer to this question is very similar to the one offered for question D5 on the (slight) superiority of females on ToM tasks. First and foremost, females’ tendency to ruminate more is statistically small; it would be inaccurate to state that females excessively ruminate and that males do not ruminate. Again, these gender differences might emerge from socialization practices, where females are reinforced when focusing on their emotions and expressing them more spontaneously than males. As seen in the answer to question D5, it is possible that socialization favors verbal activity in females, which in turn could mean more inner speech use; the latter has been shown to be associated with both self-reflection [6] and self-rumination [123]—that is, we talk to ourselves both about positive and negative self-dimensions, respectively.

It is odd that females show a small advantage over males for ToM and yet slightly self-ruminate more: recall that self-rumination impedes ToM (question A3). No known explanations have been offered pertaining to this paradox.

(3) The paper states that private self-focus is more strongly associated with depression, whereas public self-focus is more strongly related to social anxiety. Why is that?

Private self-focus occurs when we think about non-visible psychological self-aspects such as our thoughts, values, goals, memories, and emotions. This thinking often initiates a comparison between the real self (e.g., “I am selfish”) and the ideal self (“I would like to be generous”), which uncovers a discrepancy between the two self-representations [124]. The admission of this discrepancy is emotionally painful and basically signifies that one is falling short of the person one wishes to be. In the long run, the accumulation of multiple discrepancies may lead to depression. Public self-focus entails thinking about visible self-aspects such as behavior, performance, and appearance. These self-aspects are open to public scrutiny, and possibly evaluation and judgment when they do not meet others’ expectations. In simple terms, public self-focus includes others’ potentially negative opinions of the self, which may cause social anxiety.

(4) Why do we experience negative affect? Should we try to eliminate unpleasant emotions altogether?

Negative affect is intrinsic to human nature and serves the very important function of signaling that something is wrong and should be addressed. Similarly to low self-esteem (see question K4), painful emotions act as a red flag that motivates self-improvement. A person who feels bad for hurting another person’s feelings might consider making amends (e.g., expressing sorrow to the target person) in order to reduce negative affect. A person who feels distraught about an embarrassing public speech might want to practice and improve public speaking to handle the upsetting emotions. To put it simply, we need negative affect to motivate change [125]. Thus, trying to eliminate unpleasant emotions would be ultimately detrimental. Note that the above does not apply to clinical depression, where the patient experiences excessive negative affect unconnected to reality; in this case, suffering serves no constructive function and should be reduced as much as possible.

(5) How can we reduce self-rumination?

Clearly, there is no known universal “cure” to self-rumination (or depression for that matter). If there was one, we would all know about it and the person or group responsible for its development would be very rich indeed. Multiple websites offer questionable techniques, such as identifying the source of your rumination and naming it, thinking about the worst-case scenario, and scheduling a worry break—these practices will *increase* rather than reduce rumination!

Two main approaches show promise: distraction and self-distancing. With distraction (e.g., reading, listening to music, and sports activities), the ruminating person is encouraged not to think about troublesome self-related elements (e.g., [126]). With self-distancing (see [127]), the ruminating individual is motivated to “discuss” negative emotions as if they were another person, very much like when one offers advice to a friend. Such advice is more likely to be detached and objective, rather than subjective and biased. One effective way to achieve self-distancing is via inner speech addressed to the self as “you” (“Why are *you* feeling that way?”) or using one’s name (“Why is *John* feeling that way?”), as opposed to “I”, which instead increases self-immersion (see [128]). Another efficient strategy is to engage in mindfulness, which comprises “decentering” [14], by observing oneself from a third-person perspective and adding a distance between the self and its inner experiences (recall the answer to question B1).

(J) Culture

(1) Are there cultural differences in ToM?

As a reminder, individualistic cultures (e.g., USA, Canada, UK) emphasize maintaining their independence from others by discovering and expressing their unique inner attributes, whereas collectivist cultures (e.g., Japan, Mexico, the Philippines) relate more to others by attending them, fitting in, and experiencing a harmonious interdependence with them [25]. Before addressing the ToM question, it is worth noting that several studies have shown that brain activation during self-reflection tasks slightly differs between individualistic and collectivist participants. To illustrate, Zhu and colleagues [129] used functional magnetic resonance imaging (fMRI) to measure brain activity when volunteers judged whether personality trait adjectives described the self, their mother, and a public person. The medial prefrontal cortex (MPFC) and anterior cingulate cortex (ACC) showed greater activation in self- than in other-judgment conditions for both individualistic and collectivist participants. Interestingly, relative to other-judgments, mother-judgments activated the MPFC in collectivist but not in individualistic participants, possibly signifying that information about the mother is closely related to the self in the former. This suggests that culture modulates the psychological structure of the self, even at the brain level (for additional examples, see [130]), and produces two different types of self-representation.

Going back to ToM, one would expect collectivist cultures to show higher ToM because of their general focus on others, including their mental states, and the opposite in individualist cultures. Without being exhaustive, one study showed no striking cultural differences in the frequency or quality of ToM [131], yet another study reported higher cognitive ToM in Iranian participants compared to American participants [132]. Thus, this remains an open question. To our knowledge, content differences in ToM (i.e., what one is thinking when thinking about others’ mental states) have not yet been investigated, but we can speculate. Given the distinct social orientations of collectivist and individualistic cultures, we could predict that the former would engage more in cooperative ToM (e.g., “He must feel sad about this, how can I help?”) and thoughts about others’ opinion of the self pertaining to norm adherence (e.g., “I hope I did the right thing in this situation and that they think I am okay”), whereas the latter would produce more competitive ToM (e.g., “Is she going to apply to the same job position? If yes, how can I gain an advantage over her?”). Clearly, these ideas await empirical testing.

(2) Are there cultural differences in inner speech?

This question remains understudied, but in an ongoing research project, Brinthaupt and colleagues [133] are compiling and summarizing the results from several countries (e.g., Turkey, Korea, Japan, Poland), where inner speech is was measured using the Self-Talk Scale (STS [92]). The STS consists of four subscales: self-reinforcement (e.g., “I talk to myself when I need to boost my confidence that I can do something difficult”), social assessment (e.g., “I talk to myself when I review things I’ve already said to others”), self-criticism (e.g., “I talk to myself when I feel discouraged about myself”), and self-management (e.g., “I talk to myself when I’m mentally exploring a possible course of action”).

Some general predictions are as follows. Past research suggests that collectivist cultures engage in more self-criticism compared to individualistic cultures (e.g., [134,135]). Thus, it is likely that collectivist groups will score significantly higher on the self-criticism subscale of the STS. Past research also shows that self-enhancement (i.e., making oneself feel better about oneself) is more common in individualistic cultures, whereas group enhancement is more prevalent in collectivistic cultures (e.g., [136]); in addition, self-promotion (i.e., highlighting one’s accomplishments) is more frequent among those with an independent focus rather than a collectivist focus [137]. Thus, we predict that individualistic persons will score higher on the self-reinforcement subscale of the STS. For self-management (i.e., planning, goal setting, and self-regulation), expectations are more difficult to formulate, although one could argue that both collectivist and individualist cultures equally need to engage in self-management. Thus, we foresee no significant score differences between the two cultures on the self-management subscale of the STS. Social assessment includes paying attention to the reactions of others, monitoring others, and engaging in ToM. There are good reasons to believe that collectivist groups will score higher on the self-management subscale of the STS compared to individualistic groups, given a greater concern for relationship harmony among the former compared to the latter, who report more interest in individual thoughts and feelings (e.g., [138]).

(3) Are there cultural differences in self-recognition?

Yes. A general pattern is that children from collectivist cultures tend to self-recognize in front of a mirror later or not at all compared to those from individualistic cultures. To illustrate, 18- to 22-month-old Vanuatu toddlers from a small island in the South Pacific passed the mirror test at significantly lower rates compared to their Canadian counterparts [139]. Somewhat more striking, compared to children from the US and Canada, children from Kenyan, Fiji, Saint Lucia, Grenada, and Peru displayed an absence of spontaneous self-directed behaviors toward a mirror [140] (for experimental variations, see [141]). Possible explanations for these observations include the following: (a) the likelihood that for some reason, mothers from collectivist cultures may not use as many verbal prompts in front of the mirror with their kids (e.g., “Who’s there in the mirror?”) compared to mothers from individualistic groups; (b) the idea that, unlike individualistic kids, those from a collectivist background are unsure of what an acceptable response in front of a mirror may be, and therefore refrain from looking at or touching themselves. These two accounts signify that children from collectivist cultures may not fail at self-recognition per se, but rather are raised in an environment less conducive to passing the mirror test.

(4) Would someone who moved to a different culture in adulthood begin to adopt aspects of the new cultural self and lose the original one?

The person would adopt new cultural values and beliefs consistent with the new culture, but would not lose his/her former cultural identity. As a matter of fact, the person could switch between the two cultural selves when moving back and forth to and from different cultural environments. This is labeled “frame switching” [142]; we can have “multicultural selves”, and we may experience multiple cultures by shifting our cultural belief systems as a function of the cultures we currently are in. In other words, new cultural self-representations can coexist with older ones in memory.

Imagine Angelo, who was born in Canada to Filipino parents. His parents have been living in Canada for a long time and display mostly individualistic characteristics. At the age of 16, Angelo travels to the Philippines for a month to visit his grandparents and other family members, who have been living all their lives in a collectivist environment. By the end of the month, Angelo will have adopted and exhibited typical collectivist qualities, such as being family-oriented, comfortable spending time with many people in small quarters, and having respect for the elderly. Angelo then flies back to Canada, and within a short period of time, switches back to his more individualistic self, seeks privacy and personal space, focuses more on his personal goals, and analyzes what is unique about him. Angelo did not lose his individualistic traits when traveling to the Philippines—he added a new cultural identity that will remain dormant when in Canada, but which will be reactivated when he goes back to his ancestral land.

(K) Self-esteem

(1) The article states that self-esteem peaks at ages 50 to 60 and then drops—why?

Self-esteem refers to how one globally evaluates oneself [143]. By the age of 50 to 60, most people have reached full performance at work, have developed a tight and large social network, and are still in fairly good health.These factors make them feel good about themselves; hence, they have relatively high self-esteem. There is individual variability of course, but past 50–60 is when most people retire, leading to less work satisfaction and income, as well as the loss of several professional relationships. This also represents a time when health issues are more likely to emerge, and when physical (e.g., vision, motor skills) and cognitive (e.g., memory) competencies start to steadily decline. This is why self-esteem also declines. It is important to note that the lowering of self-esteem in older age is not drastic. It would be inaccurate to state that older people have low self-esteem; rather, self-esteem is a bit lower compared to its level a couple of decades earlier.

(2) Is self-esteem connected to personality?

Absolutely. High self-esteem is associated with higher E (extraversion), A (agreeableness), C (conscientiousness), and O (openness to experience), as well as lower N (neuroticism; anxiety) [144]. The opposite patterns are observed for low self-esteem.

(3) What are some of the factors that shape self-esteem?

In essence, past and current life experiences that make people feel good about themselves will increase self-esteem, and vice versa. Among these, one can list past success and failure experiences, caregivers’ patterns of reinforcement and punishment toward the child, physical appearance, self-evaluations from social comparisons, acceptance or rejection from others, and competencies (see [145]).

(4) What are some consequences of low self-esteem?

As seen with self-knowledge (question B4), ToM (question D2), and self-regulation (question E2), deficits in self-processes can have dramatically negative consequences, and self-esteem is no exception. These include behavioral (e.g., crime), physical (e.g., substance abuse) and mental health problems (e.g., anxiety and depression), marital and friendship conflicts, difficulties at school and work, and thus overall lower levels of life satisfaction [26,146]. The main reason for this is that people who evaluate people themselves negatively will likely be ridden by self-doubts, which will impede decision making, self-regulation, and performance in a wide array of life outcomes. Clearly, high self-esteem is associated with the opposite outcomes.

### 1.4. Key Messages

In this section, I summarize the key lessons emerging from the answers proposed to the weekly student questions examined above. The key lessons are as follows:All self-processes are interconnected;Self-terms must be properly defined and are not interchangeable;Language and inner speech play an important role in self-processes;Controversies are numerous;The self represents a social construct;There are gender and cultural differences in self-processes;Measurement issues abound;Psychological disorders affect self-processes;Common brain activations are reported for several self-processes;Self-processes are related to personality;Deficits in some self-processes can have devastating effects;There are many unknowns.

A running theme while answering the questions is that self-processes are intimately connected. This is evident in Figure 1 from Morin [13], where the initiator of all self-processes is self-awareness, made up of self-reflection (healthy outcomes) and self-rumination (dysfunctional outcomes). Figure 1 also shows that both self-awareness processes are articulated via inner speech, which is another important message. Self-processes that are shaped by self-reflection include self-description, self-evaluation, self-concept, self-knowledge, MTT, self-regulation, ToM, and healthy self-esteem; self-criticism, self-doubt, self-blame, low self-esteem, and self-escape are generated by self-rumination. As an example of the tight relationships between self-processes, prospection depends on autobiography, and ToM, as well as self-regulation that requires self-reflection.

A corollary of the above is the observation that self-process deficits can lead to severe negative consequences. As a case in point, self-regulatory issues may cause addiction, crime, or financial problems. This also reminds us that several psychological disorders result in impeded self-processes, including BPD in self-concept confusion, depression, and anxiety in self-rumination (and vice versa). Another significant lesson is that self-processes need to be carefully defined and are not to be used interchangeably. For example, self-talk must not be equated with inner speech, as the former refers to both inner and outer self-directed speech, whereas the latter uniquely designates internal speech.

Answering the weekly student questions also made it clear that self-processes are difficult to measure and that several existing operationalizations are problematic. To illustrate, prospection can be apprehended using questionnaires, but their retrospective nature casts doubt on the accuracy of the self-reports obtained. In addition, although some claim that self-recognition represents a valid measure of private self-awareness, others are skeptical and suggest that it only assesses bodily self-awareness. This points to two other key lessons: multiple controversies are common, and peer-reviewed published articles, despite their supposed scientific rigor, may present questionable conclusions. It is still unclear if mindfulness genuinely facilitates self-knowledge, and there are good reasons to doubt so. Claiming that fish are self-aware in a prestigious journal [113] does not guarantee that this is true [114].

Connected to the above, there are still many unknowns: which self-processes, and to what extent, do animals possess, and why is it that females are more prone to self-rumination? This last question illustrates one more key lesson: there exist important individual, gender, and cultural differences pertaining to self-processes.

## 2. Conclusions

Based on the course “The Self” designed and taught by this author, student questions generated from weekly readings were presented and tentatively answered in light of currently available information pertaining to self-processes. The themes explored included central aspects in the study of the self, such as MTT, ToM, inner speech, self-knowledge, self-recognition, self-esteem, and culture. The main goal was to extract key lessons from questions that matter to students, including the inter-connectiveness of self-processes, the importance of definitions, assessment difficulties, and controversies.

The weekly questions formulated by students were certainly influenced by the course format and focus. One could argue that different questions reflecting other concerns or interests would have emerged in a course taught by another instructor trained in an alternate discipline with dissimilar views of the self. Although to our knowledge, few university courses exist that uniquely focus on the self for an entire semester; therefore, future research could compare their structure, focus, goals, assessment methods, and so on, to determine if students would raise questions of a different nature when learning in other pedagogical environments.

The self is not only central to multiple scientific areas of inquiry (e.g., psychology, psychiatry, neuroscience, anthropology, sociology) but also to human existence per se. Thus, studying the self and trying to understand its nature, structure, content, and functions is imperative. We suggest that those studying the self can learn valuable lessons from those enrolled in courses on the self: their students.

## Figures and Tables

**Figure 1 behavsci-13-00525-f001:**
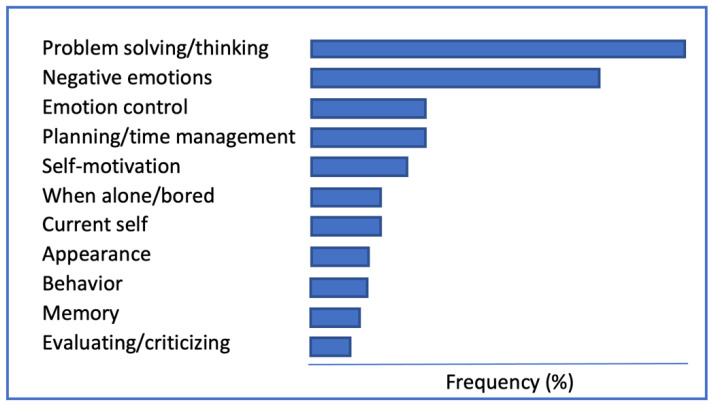
Most frequently self-reported inner speech instances (adapted from [9]).

**Table 1 behavsci-13-00525-t001:** Topics and readings in The Self course.

Topic	Readings
Self-awareness: Definitions, measures, effects, functions, antecedents, neuroanatomy, and importance of inner speech	[11,12,13]
Self-knowledge: How, and up to what point, do people know themselves? Role of mindfulness in self-knowledge	[14]
Mental time travel: Autobiography and future-oriented thoughts	[15]
Theory-of-Mind: Thinking about others’ thoughts	[16,17]
Self-regulation: Setting and reaching goals, changing the self	[18]
Self-cognition: Inner speech	[19,20]
The self in the brain: Traumatic brain injury	[21]
Self-recognition: Are non-human animals self-aware?	[22,23]
Psychopathology of the self: Rumination and negative affect	[24]
Self and culture	[25]
Self-esteem: How do you feel about yourself?	[26]

**Table 2 behavsci-13-00525-t002:** Representative inner speech instances less frequently self-reported (adapted from [9]).

Meeting people Relationships Intimate partner Social situations Family Friends Health Preferences/motives/ideas Finances Others’ opinion of self	Self-censorship Spatial orientation Rehearsal Coping Concentration Mind wandering Problems Mental simulation Rationalization Reasons for things When walking	When taking transit Leisure Physical recreation School Sleep Performing hygiene Work Education Food Immediate environment Life in general

## Data Availability

This article does not present original data.

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
