# Peer review of "The Self Course: Lessons Learned from Students’ Weekly Questions"

_behavsci, 2023, doi:10.3390/bs13070525_

Round 1
Reviewer 1 Report
It is scientific research in which the objective, methodology, and findings are interconnected. The level of content organisation is acceptable. The author has an in-depth knowledge of prior research on the same topic, applies relevant sources, and uses concepts, methods, and terminology accurately.
Author Response
Reviewer 1:
It is scientific research in which the objective, methodology, and findings are interconnected. The level of content organization is acceptable. The author has an in-depth knowledge of prior research on the same topic, applies relevant sources, and uses concepts, methods, and terminology accurately.
REPLY: I greatly appreciate the support offered by Reviewer 1. No changes to the manuscript were requested.

Reviewer 2 Report
This is an interesting paper reflecting the 9-year author’s experience of teaching the “Self” course.
The paper presents coherent description of the course, the typical topics arise and answers to students’ questions based on the relevant scientific literature. This study will be, definitely, interesting and valuable to professors teaching related courses and to wider audience.
However, to my opinion, the manuscript looks more like a methodological manual than a research paper.
I would recommend the paper for publication in the journal in case it doesn’t conflict with the journal format.
Also, there are some suggestions:
Lines 12-13. <…> The students’ weekly questions and their answers highlight what 11 is currently know about the self. <…>
“is currently know” should be replaced with “is currently known”.
Table 1 should be better formatted – bullet list on the right is not needed.
Table 3 is not necessary; bullet list might be suitable.
Lines 253-254. Although healthy (Trapnell & Campbell, 1999), self-reflection allows critical self-evaluation and mental time travel (i.e., thinking about the past and the future, not just the present).
It looks like the first part of the sentence – “Although healthy” – is not needed.

Author Response
Reviewer 2:
This is an interesting paper reflecting the 9-year author’s experience of teaching the “Self” course.
The paper presents coherent description of the course, the typical topics arise and answers to students’ questions based on the relevant scientific literature. This study will be, definitely, interesting and valuable to professors teaching related courses and to wider audience.
However, to my opinion, the manuscript looks more like a methodological manual than a research paper.
I would recommend the paper for publication in the journal in case it doesn’t conflict with the journal format.
Also, there are some suggestions:
Lines 12-13. <…> The students’ weekly questions and their answers highlight what 11 is currently know about the self. <…> “is currently know” should be replaced with “is currently known”. DONE
Table 1 should be better formatted – bullet list on the right is not needed. DONE
Table 3 is not necessary; bullet list might be suitable. DONE, inserted directly in text
Lines 253-254. Although healthy (Trapnell & Campbell, 1999), self-reflection allows critical self-evaluation and mental time travel (i.e., thinking about the past and the future, not just the present). It looks like the first part of the sentence – “Although healthy” – is not needed. I prefer to leave it: “Although healthy” is contrasted with critical self-evaluation…
REPLY: I greatly appreciate the support offered by Reviewer 2. Changes to the manuscript were requested and made accordingly (see above in bold).

Reviewer 3 Report
This study proposed about the process of "the self" with many protocols in a unique way. However, as a paper of this journal in psychology and behavioral science filed, many necessary information are not sufficiently explained, such as a review of previous research, the originality of this research, or the description of subjects. I, therefore, cannot positively evaluate this paper.
Author Response
Reviewer 3:
This study proposed about the process of "the self" with many protocols in a unique way. However, as a paper of this journal in psychology and behavioral science filed, many necessary information are not sufficiently explained, such as a review of previous research, the originality of this research, or the description of subjects. I, therefore, cannot positively evaluate this paper.
REPLY: Reviewer 3 does not recommend changes to the manuscript and rather suggests that “… I cannot positively evaluate this paper”, which either means that (s)he literally cannot assess it, or that its evaluation is negative, in which case I take it that Reviewer 3 does not endorse publication.
In defence of my submission, Reviewer 3 likely did not appreciate that the current submission does not represent a typical empirical article with review of literature, hypotheses, methods section, results, and discussion. Rather, it introduces the course The Self, explains its structure and goals, and tentatively answers a sample of students’ weekly questions. Doing so allows for the identification of the most important self-related processes as well as a discussion of how they interact – precisely what the Special Issue "Conceptual and Empirical Connections between Self-Processes" seeks.
Reviewer 3 suggests that “… many necessary information are not sufficiently explained, such as a review of previous research, the originality of this research, or the description of subjects”. I submit that a lot of information, including prior research, is presented, but not in the typical fashion in Introduction, but rather gradually, as students’ weekly questions are answered. The originality of the research is assessed both in Introduction and Conclusion, and there is no need to describe participants as this does not represent a study per se.
